Microbial communities mediating algal detritus turnover under anaerobic conditions

Morrison Jessica M. 1
Murphy Chelsea L. 1
Baker Kristina 1
Zamor Richard M. 2
Nikolai Steve J. 2
Wilder Shawn 3
Elshahed Mostafa S. 1
Youssef Noha H. noha@okstate.edu 1
1 Department of Microbiology and Molecular Genetics, Oklahoma State University , Stillwater , OK , USA
2 Grand River Dam Authority , Vinita , OK , USA
3 Department of Integrative Biology, Oklahoma State University , Stillwater , OK , USA
Eisen Jonathan
Electronic publication date: 2017 Jan 10
Publication date: 2017
Volume: 5
Electronic Location ID: e2803
Received 2016 Sep 15; Accepted 2016 Nov 18
Copyright: ©2017 Morrison et al.
Copyright year: 2017
Copyright holder: Morrison et al.
License: This is an open access article distributed under the terms of the Creative Commons Attribution License, which permits unrestricted use, distribution, reproduction and adaptation in any medium and for any purpose provided that it is properly attributed. For attribution, the original author(s), title, publication source (PeerJ) and either DOI or URL of the article must be cited.
License URL: https://creativecommons.org/licenses/by/4.0/

Keywords: Algal detritus, Anaerobic degradation, Enrichment

Funding: National Science Foundation Microbial Observatories program MCB_0240683 Oklahoma State University Start-up fund This work has been supported by the National Science Foundation Microbial Observatories program (grant no. MCB_0240683), and the Oklahoma State University Start-up fund (NY). The funders had no role in study design, data collection and analysis, decision to publish, or preparation of the manuscript.

==============================
Background

Algae encompass a wide array of photosynthetic organisms that are ubiquitously distributed in aquatic and terrestrial habitats. Algal species often bloom in aquatic ecosystems, providing a significant autochthonous carbon input to the deeper anoxic layers in stratified water bodies. In addition, various algal species have been touted as promising candidates for anaerobic biogas production from biomass. Surprisingly, in spite of its ecological and economic relevance, the microbial community involved in algal detritus turnover under anaerobic conditions remains largely unexplored.

Results

Here, we characterized the microbial communities mediating the degradation of Chlorella vulgaris (Chlorophyta), Chara sp. strain IWP1 (Charophyceae), and kelp Ascophyllum nodosum (phylum Phaeophyceae), using sediments from an anaerobic spring (Zodlteone spring, OK; ZDT), sludge from a secondary digester in a local wastewater treatment plant (Stillwater, OK; WWT), and deeper anoxic layers from a seasonally stratified lake (Grand Lake O’ the Cherokees, OK; GL) as inoculum sources. Within all enrichments, the majority of algal biomass was metabolized within 13–16 weeks, and the process was accompanied by an increase in cell numbers and a decrease in community diversity. Community surveys based on the V4 region of the 16S rRNA gene identified different lineages belonging to the phyla Bacteroidetes, Proteobacteria (alpha, delta, gamma, and epsilon classes), Spirochaetes, and Firmicutes that were selectively abundant under various substrate and inoculum conditions. Within all kelp enrichments, the microbial communities structures at the conclusion of the experiment were highly similar regardless of the enrichment source, and were dominated by the genus Clostridium, or family Veillonellaceae within the Firmicutes. In all other enrichments the final microbial community was dependent on the inoculum source, rather than the type of algae utilized as substrate. Lineages enriched included the uncultured groups VadinBC27 and WCHB1-69 within the Bacteroidetes, genus Spirochaeta and the uncultured group SHA-4 within Spirochaetes, Ruminococcaceae, Lachnospiraceae, Yongiibacter, Geosporobacter, and Acidaminobacter within the Firmicutes, and genera Kluyvera, Pantoea, Edwardsiella and Aeromonas, and Buttiauxella within the Gamma-Proteobaceteria order Enterobacteriales.

Conclusions

Our results represent the first systematic survey of microbial communities mediating turnover of algal biomass under anaerobic conditions, and highlights the diversity of lineages putatively involved in the degradation process.

Introduction

Algae represent a globally distributed group of organisms that are capable of oxygenic photosynthesis. While prevalent in aquatic marine and freshwater habitats (Cole, 1982), algal taxa are also encountered in terrestrial ecosystems such as soil, rocks, and ice/snow (Hoffmann, 1989). Collectively, algal species play an important role in global carbon, nitrogen, sulfur, and phosphorus cycling (Vanni, 2002). Taxonomically, algae are polyphyletic, and are encountered within multiple eukaryotic phyla such as the Alveolata (e.g., dinoflagellates), Stramenopiles (e.g., Bacillariophyceae, Chrysophyceae, Eustigmatophyceae), Viridiplanta (e.g., Chlorophyta), in addition to exclusively algal phyla such as the Euglenozoa, Cryptomonads, Haptophyta, and Rhodophyta (Amaral-Zettler, 2011). In addition to their complex evolutionary origin, these organisms exhibit a wide array of morphological diversity, pigments, ecological distribution, cellular composition, genome size, and cell wall structure.

A major characteristic of many algal taxa is their fast growth rate, enabling them to form conspicuous seasonal blooms under the appropriate environmental conditions. Such blooms are often associated with elevated nutrient (e.g., nitrogen and/or phosphorus) levels in the ecosystem, often resulting from anthropogenic inputs (e.g., sewage, industrial waste, and fertilizers) (Hallegraeff, 1993), as well as from destratification and nutrients resuspension (Wetzel, 2001). Classical examples of freshwater algal blooms involve members of the green algae (Chlorophyceae), whose blooms are often encountered in lakes and other freshwater habitats (Hoshaw & Mccourt, 1988), Chara blooms (commonly called Muskgrass), which seasonally occur in ponds and lakes and cause a strong and unpleasant musky odor (Durborow, 2014), as well as Diatoms, most commonly encountered in lakes (Sommer et al., 2012; Sommer et al., 1986). Fast growth is also a characteristic of many annual or perennial macroscopic taxa. The classical example of such taxa is the brown algae or Kelp (class Phaeophyceae), which is believed to be one of the most productive photosynthetic organisms and tend to attain long lengths at a very fast elongation rate (∼50–60 cm/day) (Reed, Rassweiler & Arkema, 2008).

Algae provide a large input of organic carbon into aquatic ecosystems such as coastal kelp forests (Mann, 1988), meromictic and seasonally stratified lakes (Gies et al., 2014; Xia et al., 2016; Youssef et al., 2015), and coastal areas within marine environments (e.g., the North Sea (Boon et al., 1998)). When blooming subsides, the algal detritus sinks and provides a substantial organic carbon source to microbial communities within the ecosystem (Hecky & Hesslein, 1995). Algal degradation in aquatic habitats commences at or near the water surface by the microbial phycosphere: aerobic heterotrophic bacteria that are physically attached to algal cells (Buchan et al., 2014). However, a significant fraction of algal detritus reaches the lower strata of these water bodies, providing a considerable autochthonous contribution to the carbon input in benthic layers within such ecosystems (Ask et al., 2009; Hecky & Hesslein, 1995). Sinking of algal detritus and the subsequent increase in carbon input result in the development of anoxic conditions in the lower strata and, hence, seasonal stratification. The size, intensity, and duration of these bloom-mediated anoxic zones are expected to be accentuated by future global warming trends (Paerl & Otten, 2013). Surprisingly, while a large body of research has been conducted on elucidating the microbial community composition of the algal phycosphere in the aerobic surficial marine (Amin, Parker & Armbrust, 2012; Hasegawa et al., 2007; Sapp, Wichels & Gerdts, 2007), and freshwater habitats (Bagatini et al., 2014; Cai et al., 2014; Dittami et al., 2016; Eigemann et al., 2013; Jones et al., 2013; Muylaert et al., 2002), a surprising lack of knowledge exists regarding the microbial community and patterns of algal turnover under the anoxic conditions in the lower layers of stratified water bodies.

In addition to the importance of anaerobic degradation of algal biomass to the carbon cycle in aquatic environments, the process has recently received additional attention as an integral component in algal biofuels production schemes. Direct conversion of kelp to methane (Cannell, 1990; Prabandono & Amin, 2015; Ramaraj, Unpaprom & Dussadee, 2016) has been proposed as a promising approach for biogas production (Samson & Leduy, 1982; Vanegas & Bartlett, 2013; Vergara-Fernandez et al., 2008; Wiley, Campbell & McKuin, 2011; Yen & Brune, 2007; Yuan et al., 2011). The high fat content of multiple algal taxa, e.g., Chlorella sp. (Guckert & Cooksey, 1990; Hu et al., 2008), and Chara sp. (Omer, 2013), has prompted research into their large scale production in artificial ponds, and subsequent extraction of their oil content as biodiesel (Cannell, 1990; Moazami et al., 2012; Prabandono & Amin, 2015; Ramaraj, Unpaprom & Dussadee, 2016). The economy of the process is further enhanced by anaerobic digestion of the algal detritus to produce methane as an additional source (Cannell, 1990; Prabandono & Amin, 2015; Ramaraj, Unpaprom & Dussadee, 2016) using sludge from anaerobic wastewater treatment plants as the inoculum source (Bohutskyi et al., 2015). While various aspects of the engineering and performance have been studied, there is very little documentation of the identity of the microbial community that is mediating algal detritus turnover under these anaerobic conditions.

Here, we sought to characterize patterns of algal turnover under anaerobic conditions, and identify members of the microbial community involved in the degradation of various ecologically and economically relevant algal taxa. The process was investigated in enrichments derived from three anaerobic habitats that either exhibit seasonal algal blooming, or highly eutrophic environments that receive a high input of organic compounds and previously shown to harbor a high level of microbial diversity. To our knowledge, this is the first study that systematically characterized the microbial community associated with algal degradation under anaerobic conditions.

Materials and Methods

Algal species (substrates/carbon source)

We evaluated the microbial community mediating algal detritus degradation by setting enrichments with various types of algal biomass as the only source of carbon. Three different algal species were examined: (1) Axenic Chlorella vulgaris strain UTEX 2714 (phylum Chlorophyta), representing green algae that are known to bloom during summer months (June–September) in multiple freshwater bodies within the US. Strain UTEX 2714 was obtained from the University of Texas at Austin Algal Culture collection, and cultures were maintained on proteose medium plates (composition g l−1: NaNO3, 0.25; CaCl2.2H2O, 0.025, MgSO4.7H2O, 0.075; K2HPO4, 0.075, KH2PO4, 0.175, NaCl, 0.025; proteose peptone, 1, agar, 15) at 22 ± 3 °C. Fluorescent light was used (3500 lux) on a 16:8 h light to dark cycle. Cultures were harvested by scraping the growth on the surface of agar plates, and the resulting biomass was used as the carbon source. (2) Chara sp. strain IWP: Mats of Chara (class Charophyceae) were obtained from a local pond (Innovation Way Pond in Stillwater, OK, coordinates N36°6′37.75″ W97°6′44.72″) in August 2015, and identified using morphological and microscopic analysis as Chara sp. Chara is known to grow locally in ponds in Oklahoma and peak around June–September (B Henley, pers. comm., 2016). Samples collected were thoroughly washed and soaked in DI water for 2 h, to remove other associated biomass, before they were centrifuged and the resulting biomass was used as the carbon source to represent class Charophyceae. (3) Kelp: Due to the geographical location of the study and brown algae being common occurrences in marine environments, we were not able to obtain a fresh kelp sample to be used as a substrate. Alternatively, Ascophyllum nodosum (phylum Phaeophyceae) representing brown algae was obtained as a whole dried powder from a local provider (Starwest Botanicals®, Sacramento, CA, USA) and directly used as the carbon source.

It is worth noting that, with the exception of the axenic Chlorella culture, the Chara and the kelp samples were not guaranteed axenic and a minimal input of bacteria with the carbon source in these enrichment bottles could not be ruled out.

Enrichment (inoculum) sources

Grand Lake (GL)

Samples were obtained from Grand Lake O’ the Cherokees (hereafter Grand Lake) to investigate the microbial community involved in algal turnover. The lake is a large (188 Km2) man-made lake in Northeastern OK operated by the Grand River Dam Authority, which keeps continuous records of the lake water geochemistry. During summer months (starting in June), large areas of the lake become seasonally stratified, with deeper layers (June–September) becoming completely anoxic. Within these sites, surface chlorophyl1-a concentration peaks in May–July, followed by algal biomass sinking to deeper anaerobic layer in September (Fig. S1). Sampling from the hypolimnion of Tree and Dream sites in GL occurred in September 2015 using a 4.0-L Van Dorn Bottle. Whole water samples were stored on ice until processed in the laboratory where the lake water was centrifuged under anaerobic conditions to collect biomass used as inoculum source.

Wastewater treatment plant (WWT)

While wastewater treatment reactors do not represent an algae rich habitat, the use of WWT material as an inoculum for biogas production from algal biomass sources has been gaining considerable attention (Ward, Lewis & Green, 2014). The process is justified by the high organic content and adaptation to organic matter turnover under anaerobic conditions, coupled to the ready availability of WWT inocula (Sialve, Bernet & Bernard, 2009). Various aspects of the engineering, kinetics, and the economy of the process, as well as optimization of the inoculum load and substrate load, and algae pretreatment methods have been investigated (Hlavínek et al., 2016; Mudhoo, 2012; Nabarlatz et al., 2013; Ramaraj, Unpaprom & Dussadee, 2016; Samson & Leduy, 1982; Vergara-Fernandez et al., 2008; Ward, Lewis & Green, 2014), but little research on the identity of microorganisms mediating such process has been conducted. Samples were obtained from secondary treatment sludge in the city of Stillwater, OK wastewater treatment plant in September 2015. The sample was collected anaerobically and transferred to the laboratory (5 miles away), where they were promptly centrifuged under anaerobic conditions and used as inoculum source.

Zodletone spring (ZDT)

Zodletone spring is an anaerobic surficial spring in southwestern OK (35°0′9″N 98°41′17″W). Due to the constant ejection of sulfide laden water from the spring source, the shallow spring is light exposed, yet mostly anoxic (Buhring et al., 2011). The microbial community in the spring has been extensively investigated (Coveley, Elshahed & Youssef, 2015; Youssef, Couger & Elshahed, 2010), and the spring harbors a highly diverse community of phototrophs, chemolithotrophs, and heterotrophs. Samples were collected from Zodletone spring source in August 2015 in a filled mason jar (to maintain anoxic conditions), transferred to the laboratory at 4 °C and used as inoculum source on the same day of sampling.

Enrichments setup

Enrichments were prepared in 120 ml serum bottles under anaerobic conditions. Nine different treatments (three algal substrates × three different inoculum sources) were set up in triplicates. Serum bottles contained 45 ml of an anoxic solution containing (per liter): 150 ml of minerals solution I (K2HPO4 3 g l−1), 150 ml of mineral solution II (g l−1: KH2PO4, 3; (NH4)2SO4, 6; NaCl, 6; MgSO4.7H2O, 0.6, and CaCl2.2H2O, 0.6), 10 ml Balch vitamins solution (mg l−1: biotin, 2; folic acid, 2; pyridoxine-HCl, 10; thiamine-HCl, 5; riboflavin, 5; nicotinic acid, 5; calcium pantothenate, 5; vitamin B12, 0.1; p-aminobenzoic acid, 5; lipoic acid, 5), 1 ml of Wolin’s metal solution (g l−1: EDTA, 0.5; MgSO4.6H2O, 3.0; MnSO4.H20, 0.5; NaCl, 1; CaCl2.2H2O, 0.1; FeSO4.7H2O, 0.1; ZnSO4.7H2O, 0.1; CuSO4.7H20, 0.01; AlK(SO4)2, 0.01; Na2MoO4.2H20, 0.01; boric acid, 0.01; Na2SeO4, 0.005; NiCl2.6H20, 0.003; CoCl2.6H20, 0.1). The media were amended with L-cysteine hydrochloride (0.05 g/l final concentration) as a reductant, and resazurin (0.0001% final concentration) as a redox indicator, boiled under a stream of N2 gas, dispensed in the serum bottles, autoclaved, cooled, then transferred to an anaerobic chamber (Coy Laboratory Products Inc., Ann Arbor, MI, USA) where 5 g of sediment (Zodletone, ZDT), 5 g of sludge (wastewater treatment plant, WWT), or 5 ml of concentrated lake water (the pellet obtained after centrifugation of 1.5 L of Grand Lake water (GL)) were added as the inoculum sources. Algal biomass was added as a substrate (∼0.25 g per bottle). In addition, substrate unamended controls (i.e., ZDT, WWT, and GL enrichments with no algal substrates) were included. After enrichment preparation in the anaerobic chamber, the bottles were stoppered, sealed, taken out of the chamber and the headspace in the bottles was changed by repeated flushing with 100% N2. Samples were incubated at room temperature (22 °C) in the dark. Enrichments were periodically sampled (at 4, 7, 8, and 10 weeks) for DNA extraction by thoroughly mixing the serum bottle and anoxically withdrawing 3 ml of the enrichment. At the end of the enrichment process (13 weeks for GL, 16 weeks for ZDT and WWT), bottles were sacrificed, centrifuged and 3 ml of the pellet was used for DNA extraction. The rest of the pellet was used for chemical analysis of the remaining algal detritus.

DNA extraction, amplification, and sequencing

DNA was extracted using the PowerSoil® DNA Isolation Kit (MO BIO Laboratories, West Carlsbad, CA, USA) as per the manufacturer’s instruction. DNA from triplicate treatments was pooled prior to amplification and sequencing. DNA from substrate-unamended controls was also extracted (hereafter pre-enrichment sample). The extracted and pooled DNA (n = 30; 3 inoculum sources × 3 algal substrates  × 3 time points, plus 3 pre-enrichment samples) was quantified using Qubit fluorometer (Life technologies®, Carlsbad, CA, USA). The genes for the V4 hypervariable region of 16S rRNA were amplified using the prokaryotic-specific primer pair 515F and 806R (Wang & Qian, 2009) to avoid amplification of eukaryotic 18S rRNA. Products were sequenced using paired-end Illumina Miseq platform, as previously described (Caporaso et al., 2012). Both PCR amplification and Illumina sequencing were conducted using the services of the Genomic Sequencing and Analysis Facility (GSAF) at the University of Texas at Austin. The sequences are deposited in the SRA database under accession number SRP083898.

Data analysis

Sequence processing

mothur software (Schloss et al., 2009) was used for most of the sequence processing and operational taxonomic unit (OTU) assignments. Most of the analyses were conducted on the cowboy server, a high performance super computer housed at the Oklahoma State High Performance Computing Center (https://hpcc.okstate.edu/). For quality control purposes and to eliminate poor quality sequences, an average quality score of 25 was chosen as the threshold value below which sequences were considered of poor quality and removed from the dataset. In addition, sequences that contained an ambiguous base (N), sequences having a homopolymer stretch longer than 8 bases, and sequences longer than 293 bp were also removed from the datasets.

High-quality reads were aligned in mothur using the Silva alignment database as a template. Aligned sequences were then filtered to remove columns that corresponded to ‘.’ or ‘-’ in all sequences. Filtered alignments were then subjected to a pre-clustering de-noising step using a pseudo-single linkage algorithm with the goal of removing sequences that are likely due to sequencing errors (Huse et al., 2010). Possible chimeric sequences were identified and removed using chimera.slayer in mothur. The taxonomy of the remaining sequences was identified according to the Silva taxonomic outline (Release 123, https://www.arb-silva.de/). The aligned, filtered, de-noised, and chimera-free sequences were used to generate an uncorrected pair wise distance matrix. Sequences were clustered into operational taxonomic units (OTUs) at 0.03% sequence divergence cutoff using the vsearch clustering method employed through mothur. A shared file was created and was used for subsequent analyses. Most of the above steps were derived from the MiSeq SOP available from the mothur website (http://www.mothur.org/wiki/MiSeq_SOP).

Criteria used to define lineages contributing to the degradation process

Phyla considered significant to the degradation process were empirically defined as those phyla that constituted 5% or more of the community at any time during enrichment. These include phyla that were abundant prior to enrichment and remained abundant during and after enrichment, phyla that transiently increased in abundance during part of the enrichment but then decreased in abundance by the end of enrichment, and phyla that significantly and progressively increased in abundance with enrichment time. Within these abundant phyla, genera considered significant to the degradation process were also empirically defined as those whose percentage abundance represented 1% or more of the total abundance.

Diversity and community structure comparisons

Various alpha diversity indices (Shannon, Chao, Ace, Good’s coverage) were performed on individual datasets in mothur. When comparing species richness across datasets (e.g., number of observed OTUs, species richness estimates using Chao and Ace estimators), numbers were reported per sample size to normalize for the differences in the number of sequences obtained between datasets. Beta diversity based on community structure was assessed by calculating pairwise Bray–Curtis dissimilarity indices and using the output to construct non-metric multidimensional scaling (NMDS) plots in mothur.

Statistical analyses

To study the significance of the effect of inoculum source versus algae type on community structure, we performed an analysis of variance using the function Adonis in the R statistical package vegan. The effect was visualized using the percentage abundances of significant phyla/classes (≥5% as defined above) in a constrained correspondence analysis (CCA) using the function cca in the R statistical package vegan.

Quantitative PCR

We used qPCR to quantify total Bacteria, total Archaea, as well as methanogens and sulfate-reducing bacteria in the enrichments using a MyiQ thermocycler (Bio-Rad Laboratories, Hercules, CA) and SYBR GreenER™ qPCR SuperMix for iCycler® Instrument (Life Technologies). Primer pair EUB-338F/UNI518R (Fierer et al., 2005) was used to amplify the 16S rRNA genes from the total bacterial community, primer pair A341F/A519R (Qian et al., 2011) was used to amplify 16S rRNA genes from the total archaeal community, primer pair mlas-ModF/mcrA-R (Angel, Claus & Conrad, 2012) was used to amplify the methyl-Coenzyme M reductase (mcrA) gene from the total methanogenic community, and primer pair Dsr2060F/Dsr4R (Balk, Keuskamp & Laanbroek, 2015) was used to amplify the dissimilatory sulfite reductase (dsrB) gene from the total sulfate reducing community. The 25-µl PCR reaction mixtures contained 0.3 µM of each forward and reverse primers (final concentration), 2 µl extracted template DNA, and 12.5 µl SYBR GreenER™ qPCR SuperMix. The reactions were heated at 95 °C for 8.5 min, followed by 40 cycles, with one cycle consisting of 30 s at 95 °C, 45 s at 50 °C (for total bacteria, total archaea, and methanogens) or 55 °C (for sulfate reducers), 30 s at 72 °C, and 15 s at 85 °C for signal reading. To calculate the total number of cells belonging to total bacteria, total archaea, methanogens, and sulfate reducers in the enrichments, a standard curve was generated using DNA from Bacillus subtilis strain 168 (ATCC 23857), Haloferax sulfurifontis strain M6 (DSM 16227), Methanosarcina hungatei strain JF1 (ATCC 27890), and Desulfovibrio desulfuricans strain G20 (ATCC BAA-1058), respectively. To account for the multiple copies of 16S rRNA genes per cell, the number of copies obtained from the standard curve was divided by an empirical value of 3.5 (average of 1–6 copies of rRNA genes in one cell). However, since the mcrA and dsrB genes are known to be present as single copies in methanogens, and sulfate reducers, respectively, no such adjustment of the total number of cells was required when calculating the total number of cells belonging to methanogens and sulfate reducers.

Chemical analysis of algal detritus

We studied the change in chemical composition of algal detritus during enrichment by quantifying the total soluble carbohydrates, total starch, total protein, and total lipid content of algal biomass pre and post enrichment. Algal detritus was dried overnight at 40 °C then weighed (DWf; final dry weight) and ground to fine material. The ground material was first used for protein extraction using the method described previously (Rausch, 1981). Briefly, algal detritus was extracted 2–3 times with 0.5N NaOH at 80–100 °C for 10 min followed by cooling and centrifugation to collect the total protein in the supernatant. Total protein extracts were frozen at −20 °C until assayed using Qubit Protein Assay Kit (Life technologies). The pellet remaining after protein extraction was used for extraction of total soluble carbohydrates and starch. The pellets were first washed 2–3 times with 1 ml acetone to remove pigments. Total soluble carbohydrates were then extracted from the pellet using 80% ethanol according to the protocol in Maness (2010) and the total ethanol extract was dried overnight at 40 °C followed by dissolving the dried extract in water. Total starch remaining in the pellet was extracted by boiling with 1.1% HCl for 30 min followed by centrifugation. Total soluble carbohydrates, as well as total starch extracted were quantified using the anthrone method (Maness, 2010). The total crude lipids were extracted from dried algal material with chloroform and quantified with a Nile red assay modified for microplates using the protocol described previously (Higgins et al., 2014).

Results

Sequencing output

A total of 1,007,906 sequences were obtained from all enrichments. After implementation of all quality control criteria described above, 889,230 sequences (88.2%) were retained for further analysis. The average number of sequences per dataset was 26,946. The calculated Good’s coverage for the majority of samples at putative species (OTU0.03, 30 out of 32 samples) and family (OTU0.10, 31 out of 32 samples) levels were always above 96 and 98.2% (average 98.4, and 99.5%, respectively), strongly indicating that the communities have been adequately sampled in all enrichments (Table S1).

Enrichment progress and diversity patterns

Multiple lines of evidence strongly indicate that in all nine treatments, algal detritus degradation occurred and was coupled to an increase in prokaryotic cell numbers and a decrease in alpha diversity both implying enrichment of specific taxa. Visual inspection of all enrichments revealed significant loss of the dried kelp powder, and the algal biomass (Chlorella and Chara) at the conclusion of the experiment. Final time point analysis demonstrated that the majority of the starting dry weight of Chara (86.3, 94.5, and 98.0%), Chlorella (96.0, 98.0, and 99.0%), and, to a relatively lower extent, kelp (56.7, and 33.6, 83.3%) in ZDT, WWT, and GL enrichments, respectively, was metabolized at the conclusion of the experiment. Analysis of the chemical composition of the remaining algal detritus in comparison to the starting material showed that the carbohydrate, lipid, and protein contents of the algal detritus were consumed to varying extents (Table 1). Quantitative PCR (qPCR) demonstrated a progressive increase in bacterial 16S rRNA gene copies/ml enrichment in all samples. An increase of 3.5–88.5 fold in total number of bacterial cells was observed by week 13–16 in all enrichments and 14.6–2,142 fold in the total number of archaeal cells was observed by week 13–16 in 6 out of 9 enrichments (Fig. 1). Finally, we followed the change in diversity estimates in the enrichments datasets as a proxy for enrichment progress. At the end of all enrichments (weeks 13 or 16), the number of observed OTUs0.03 and OTUs0.1 as well as the estimated species richness (using both Chao and ACE estimators) decreased compared to the pre-enrichment sample, hence indicating the selection for few taxa (Table 2).

Table 1 Percentage of various algal components consumed under different enrichment conditions.a

	ZDT enrichment	WWT enrichment	GL enrichment	
Algal detritus components	Chara	Chlorella	Kelp	Chara	Chlorella	Kelp	Chara	Chlorella	Kelp	
Carbohydrate	87	96.5	52.9	99.7	98	15.7	98.3	99.2	86.2	
Protein	72	92.9	60	94	96.9	70	96	97.6	82	
Lipid	100	NAb	71	62.5	NAb	70	96	NAb	86	
% Biomass lostc	86.3	96	56.7	94.5	98	33.6	98	99	83.3	
Notes.

a Carbohydrate, protein, and lipid contents of algal detritus were determined before and after enrichment. Percentages are calculated based on the dry weight at Tf. Original algal detritus composition was as follows (%Carbohydrate: %Protein: %Lipid): Chara, 88:6.5:5.5; Chlorella, 86.7:13.3:0; Kelp, 67.9:14.1:18.

b Lipids in Chlorella biomass were BDL.

c Based on dry weight remaining at the end of enrichment (DWf), and the initial dry weight used for enrichment (DW0) using the equation: % biomass loss = (DW0 − DWf)/DW0 × 100. Initial dry weight for kelp was equivalent to the weight added to each enrichment bottle since it was in dry powder form. However, initial dry weight for Chara and Chlorella was determined by incubating an amount equivalent to the wet weight added to each enrichment bottle overnight at 40 °C then weighing its dry weight following moisture loss.

Figure 1 Total number of bacterial, archaeal, sulfate-reducing, and methanogenic cells in the pre-enrichment sample (■) versus post-enrichment samples at week 4 for GL enrichments or week 7 for ZDT and WWT enrichment (□), post-enrichment samples at week 8 for GL enrichments or week 10 for ZDT and WWT enrichment ( ), and post-enrichment samples at week 13 for GL enrichments or week 16 for ZDT and WWT enrichment ( ) as measured by quantitative PCR.

The enrichment inoculum source is shown on the left, while the algae type used is shown on top. Error bars are averages ± standard deviations from three biological replicates. Linear regression analysis was performed to examine the trend of increase in cell numbers with the weeks of enrichment, and the significance of such trend was tested by calculating the P-values of the F-statistics obtained, where “**” denotes significant P-value < 0.05, “*” denotes p-value > 0.05 but < 0.1, “NS” denotes non-significant P-value > 0.1, and “ND” refers to cases where the linear regression analysis was not performed because two or more samples were below the detection level of the qPCR. In the few cases, denoted by a superscript letter a, where the total cell numbers increased initially then decreased by the last week of enrichment, the linear regression was only carried on total numbers from the first three weeks of enrichments.

Microbial community structure analysis

Bray–Curtis dissimilarity indices at OTU0.03 coupled to non-metric multidimensional scaling (NMDS) were used to compare and visualize differences in the microbial community structure between all enrichments at all sampled data points. At first glance, it was apparent that the enriched microbial communities (week 7-10-16 in cases of WWT and ZDT microcosms, or week 4-8-13 in case of GL microcosms) within each algae type-enrichment source combination (n = 9, blue, green, and red shapes in Fig. 2) clustered closely together, and were distinct from the pre-enrichment microbial community (black shapes in Fig. 2). This observation strongly suggests that the abundant microbial community obtained during the first few weeks of enrichment (week 4 or 7) persisted throughout the enrichment and was responsible for the algal biomass degradation observed at the end of the enrichment (Table 1). Analysis of the effect and relative contribution of algae type (Chlorella, Chara, or kelp) versus inoculum source (ZDT, WWT, and GL) on the enriched microbial communities revealed that kelp selects for a distinct and highly similar microbial community, regardless of the inoculum source (ZDT, WWT, and GL) (Fig. 2A). On the other hand, within Chlorella and Chara-derived enrichments, the inoculum source, rather than the algae type appears to be the more important factor in shaping the microbial communities (Fig. 2A). This is evident by the presence of three distinct clusters in the NMDS plot corresponding to the three sources of inoculum (ZDT, WWT, and GL) (Fig. 2A).

Table 2 Number of OTUs0.03 and OTUs0.1 normalized to the total number of sequences, and the estimated species richness (using both Chao and ACE estimators) normalized to the total number of sequences.

		Chara	Chlorella	Kelp	
Source_cutoffa	Weeks of enrichment	OTUs	Chao	ACE	OTUs	Chao	ACE	OTUs	Chao	ACE	
ZDT_0.03	0	0.193	0.373	0.501	0.193	0.373	0.501	0.193	0.373	0.501	
	7	0.021	0.040	0.056	0.031	0.054	0.067	0.060	0.130	0.255	
	10	0.030	0.051	0.068	0.022	0.043	0.058	0.069	0.148	0.265	
	16	0.028	0.059	0.078	0.025	0.051	0.069	0.046	0.118	0.208	
ZDT_0.1	0	0.051	0.078	0.088	0.051	0.078	0.088	0.051	0.078	0.088	
	7	0.008	0.011	0.010	0.016	0.023	0.028	0.026	0.043	0.058	
	10	0.012	0.017	0.016	0.009	0.013	0.015	0.026	0.042	0.053	
	16	0.011	0.016	0.019	0.009	0.012	0.014	0.019	0.029	0.040	
GL_0.03	0	0.043	0.091	0.140	0.043	0.091	0.140	0.043	0.091	0.140	
	4	0.008	0.019	0.033	0.014	0.025	0.035	ND	ND	ND	
	7	0.014	0.032	0.044	0.010	0.020	0.024	0.092	0.116	0.123	
	13	0.022	0.055	0.057	0.011	0.018	0.029	0.036	0.063	0.076	
GL_0.1	0	0.020	0.032	0.042	0.020	0.032	0.042	0.020	0.032	0.042	
	4	0.003	0.007	0.009	0.008	0.015	0.022	ND	ND	ND	
	7	0.006	0.015	0.020	0.005	0.009	0.010	0.051	0.058	0.061	
	13	0.010	0.015	0.014	0.005	0.007	0.008	0.017	0.026	0.029	
WWT_0.03	0	0.048	0.087	0.108	0.048	0.087	0.108	0.048	0.087	0.108	
	7	0.043	0.072	0.092	0.066	0.109	0.140	0.013	0.039	0.055	
	10	0.020	0.036	0.046	0.033	0.052	0.061	0.020	0.043	0.064	
	16	0.030	0.051	0.068	0.021	0.036	0.043	0.012	0.025	0.035	
WWT_0.1	0	0.013	0.022	0.029	0.013	0.022	0.029	0.013	0.022	0.029	
	7	0.018	0.025	0.024	0.030	0.043	0.048	0.006	0.009	0.012	
	10	0.006	0.009	0.009	0.013	0.019	0.019	0.006	0.010	0.014	
	16	0.011	0.015	0.017	0.008	0.010	0.010	0.004	0.007	0.009	
Notes.

a Source refers to the inoculum source, while cutoff refers to the percentage divergence cutoff used to assign sequences into operational taxonomic units (OTUs). For each inoculum source, the numbers are shown for OTUs at the putative species level (0.03) and the putative order level (0.1).

ND Not determined due to the small number of sequences obtained for this dataset

In addition, analysis of variance (using Adonis function) showed that both the algae type and the source of inoculum were significant in shaping the microbial community albeit to varying levels (p-value for algae type = 0.028, p-value for inoculum source = 0.001). To decipher the relative contributions of algae type versus inoculum source on the microbial community composition at the phylum/class level we employed canonical correspondence analysis (CCA) using the enriched phyla/classes relative abundances. The results (Fig. 2B) confirmed the above observation, where the algae type appears to have shaped the microbial community in case of kelp enrichments, while within Chara and Chlorella enrichments, the source of inoculum played a more important role in shaping the community (Fig. 2B).

Phylogenetic affiliation of enriched taxa in algal enrichments

In general, a handful of phyla were consistently abundant across all treatments and were considered significant to the algal degradation process (see the criteria we used for defining such phyla in ‘Materials and Methods’). These phyla were: Firmicutes (in all nine enrichments), Bacteroidetes (in six enrichments), Spirochaetes (in five enrichments), and the Gamma (5 enrichments), Delta (6 enrichments), Alpha, Beta, and Epsilon (one enrichment) Proteobacteria (Figs. 3–5). However, within this limited number of phyla, the family/genus level enrichment patterns varied widely, suggesting the involvement of a wide range of bacterial lineages in the degradation process. Below, we provide a detailed analysis of the enriched families/genera across various enrichments. The detailed microbial community composition across all datasets is shown in Table S2.

Chara microcosms

In Chara microcosms, Bacteroidetes, Firmicutes, and Delta-Proteobacteria were consistently abundant (Table 3) and, collectively, constituted the majority (40.2% to 72.7%) of the community at the end of enrichment. Spirochaetes were abundant only in WWT and ZDT enrichments, while Gamma-Proteobacteria were abundant only in GL and ZDT enrichments (Figs. 3–5).

Figure 2 Microbial community structure analysis in the enrichment microcosms (n = 26) as compared to the pre-enrichment inoculum sources (n = 3).

The inoculum sources are denoted by shapes; ZDT (●), WWT (⬣), and GL (■), and the algae types are denoted by color; Chara (blue), Chlorella (green), Kelp (red), and no algae, i.e., pre-enrichment community, (black). Each enrichment condition (inoculum source × algae type) is represented by 3 sample points corresponding to the weeks during enrichment, except for GL-kelp enrichment where the dataset from week 4 is not shown due to the small number of sequences obtained with this dataset. (A) Non-metric multidimensional scaling plots based on Bray–Curtis dissimilarity indices at the species level (0.03). For Chara and Chlorella enrichments, communities grouped by the inoculum source, while Kelp enrichments grouped by the algae type. (B) Canonical correspondence analysis using the abundant phyla/classes relative abundances to study the effect of algae type and inoculum source on the microbial community composition. Here, the same pattern is observed at the phylum/class level, where the community structure of Chara and Chlorella enrichments were similar and grouped by inoculum source, while the microbial community of Kelp enrichments were quite distinct and grouped together regardless of the inoculum source. This pattern is reflected on the direction of the factors arrows, where the algae type is pointing in the direction of the Kelp enrichments. The CCA also depicts the abundant phyla/classes that seem to shape the microbial community in the different enrichments; Gamma-Proteobacteria in GL Chara and Chlorella enrichments, Spirochaetes and Firmicutes in ZDT-Chlorella enrichment, Delta-Proteobacteria and Bacteroidetes in ZDT-Chara enrichments and WWT Chara and Chlorella enrichments, and Epsilon-Proteobacteria and Firmicutes in Kelp enrichments regardless of the inoculum source. The constrained variables explained 57% of the variance.

Figure 3 Microbial community composition in ZDT enrichments.

Abundant phyla/classes are shown as area charts for Chara (i), Chlorella (ii), and Kelp (iii) enrichments for each inoculum source. Phyla that constituted 5% or more of the community at any time during enrichment were considered significant to the degradation process and are shown in the area charts. These include phyla that were abundant prior to enrichment and remained abundant during and after enrichment (e.g., Bacteroidetes in Chara and Chlorella enrichments (i, ii)), and phyla that significantly and progressively increased in abundance with enrichment time (e.g., Firmicutes in Kelp enrichments (iii)). Bar charts depict the relative abundance of abundant genera (>1%) in each of the abundant phyla/classes shown in i-ii-iii. These include Proteobacteria (iv), Bacteroidetes (v), Firmicutes (vi), and Spirochaetes (vii). The X-axis denotes the weeks of enrichment (i–iii), or the weeks of enrichment and algae type (iv–vii). “0” denotes the community composition in the pre-enrichment inoculum source.

Figure 4 Microbial community composition in WWT enrichments.

Abundant phyla/classes are shown as area charts for Chara (i), Chlorella (ii), and Kelp (iii) enrichments for each inoculum source. Phyla that constituted 5% or more of the community at any time during enrichment were considered significant to the degradation process and are shown in the area charts. These include phyla that were abundant prior to enrichment and remained abundant during and after enrichment (e.g., Bacteroidetes in Chara and Chlorella enrichments (i, ii)), phyla that transiently increased in abundance during part of the enrichment but then decreased in abundance by the end of enrichment (e.g., Delta-Proteobacteria in Chara and Chlorella enrichments (i, ii)), and phyla that significantly and progressively increased in abundance with enrichment time (e.g., Firmicutes in Kelp enrichments (iii)). Bar charts depict the relative abundance of abundant genera (>1%) in each of the abundant phyla/classes shown in i-ii-iii. These include Proteobacteria (iv), Firmicutes (v), Bacteroidetes (vi), and Spirochaetes (vii). The X-axis denotes the weeks of enrichment (i–iii), or the weeks of enrichment and algae type (iv–vii). “0” denotes the community composition in the pre-enrichment inoculum source.

Figure 5 Microbial community composition in GL enrichments.

Abundant phyla/classes are shown as area charts for Chara (i), Chlorella (ii), and Kelp (iii) enrichments for each inoculum source. Phyla that constituted 5% or more of the community at any time during enrichment were considered significant to the degradation process and are shown in the area charts. These include phyla that were abundant prior to enrichment and remained abundant during and after enrichment (Gamma-Proteobacteria in Chara and Chlorella enrichments (i, ii)), and phyla that significantly and progressively increased in abundance with enrichment time (e.g., Firmicutes in Kelp enrichments (iii)). Bar charts depict the relative abundance of abundant genera (>1%) in each of the abundant phyla/classes shown in i-ii-iii. These include Bacteroidetes (iv), Firmicutes (v), Delta and Epsilon-Proteobacteria (vi), Gamma-Proteobacteria (vii), Alpha and Beta Proteobacteria (viii), and Planctomycetes (ix). The X-axis denotes the weeks of enrichment (i–iii), or the weeks of enrichment and algae type (iv–ix). “0” denotes the community composition in the pre-enrichment inoculum source.

Within the Bacteroidetes, the uncultured putative genus VadinBC27 was consistently enriched (Table 3) regardless of the inoculum source. This uncultured subgroup within the order Bacteroidales has been previously identified as a major lineage in anaerobic digestors (Liu et al., 2016; Riviere et al., 2009; Xie et al., 2014; Xu et al., 2012) and was implicated as an anaerobic fermenter of sludge or other carbon sources. Other enriched Bacteroidetes members include the genera Mangroviflexus (ZDT microcosms), previously identified as an important in-situ fermenter of organic matter-rich soil (Ding et al., 2016) and anaerobic cellulolytic microcosms (Gao, Xu & Ruan, 2014), Paludibacter (WWT and ZDT microcosms), previously enriched from anaerobic freshwater sediment (Sanchez-Andrea et al., 2013) and shown to be an anaerobic propionate-producer (Qiu et al., 2014; Ueki et al., 2006), Bacteroides (WWT and GL microcosms), a well-documented complex carbohydrate degrader in a wide range of environments (Adamberg et al., 2015; Dongowski, Lorenz & Anger, 2000; Jiménez, Chaves-Moreno & Van Elsas, 2015), Barnesiella (GL microcosms), a known fermentative gut microbe (Wang et al., 2015), and WCHB1-69 (ZDT microcosms), a yet-uncultured Bacteroidetes family previously encountered in organic solvent-contaminated aquifers and anaerobic digestors (Dojka et al., 1998; Xu et al., 2012) (Table 3 and Figs. 3–5).

Within the Firmicutes, all enriched taxa belonged to the order Clostridiales, a ubiquitous order of strictly anaerobic, fermentative bacteria (Xia et al., 2015). However, the profile of enriched families/genera within this order depended on the inoculum source. Members of Ruminococcaceae were abundant in all microcosms, while members of the families Clostridiaceae_1 and Family XIII were enriched only in ZDT microcosms, and members of the Lachnospiraceae and Veillonellaceae were enriched only in GL microcosms (Table 3 and Figs. 3–5).

Within the Delta-Proteobacteria, the sulfate-reducing genera Desulfovibrio, Desulfobacter, Desulfobulbous, and Desulfomicrobium were encountered as predominant members in enrichments from some or all inoculum sources. Enrichment of sulfate reducers in ZDT and WWT microcosms was accompanied by a significant decrease in the amount of sulfate in the enrichments (Fig. S2). On the other hand, sulfate concentration did not decrease in GL microcosms (Fig. S2) in spite of the apparent enrichment of SRBs (6.9% of the total enriched taxa). Similar results were previously shown for members of Desulfovibrio and Desulfobulbous in biofilms (Santegoeds et al., 1998), where not all SRBs detected by culture-independent techniques were found to be sulfidogenically active.

Members of the Spirochaetes were enriched in WWT and ZDT microcosms. The genus Spirochaeta and the yet uncultured family SHA-4 were identified as the major enriched Spirochaetes members in both enrichments. Both lineages appear to be widely distributed in a wide array of freshwater and marine habitats and enrichments (Bozo-Hurtado et al., 2013; Gu et al., 2004; Leschine, Paster & Canale-Parola, 2006; Wang et al., 2014).

Members of the Gamma-Proteobacteria were enriched in ZDT and GL microcosms. However, the identity of enriched families/genera differed depending on the inoculum source, where Kluyvera and unclassified Enterobacteriaceae were enriched in ZDT microcosms, while Buttiauxella, Pantoea and Aeromonas were enriched in GL microcosms. All such members are known carbohydrate fermenters previously encountered in microbial consortia degrading plant biomass (Jiménez, Chaves-Moreno & Van Elsas, 2015; Jiménez et al., 2016), in earthworm gut enrichments (Wust, Horn & Drake, 2011), and in microbial mats from bicarbonate- and ferrous-iron-rich spring (Hegler et al., 2012).

Chlorella microcosms

Enrichment patterns in Chlorella microcosms were very similar to Chara enrichments; with the phyla Bacteroidetes, Firmicutes, and Delta-Proteobacteria consistently enriched in microcosms derived from all three inoculum sources (ZDT, WWT, and GL), Spirochaetes only enriched in WWT and ZDT microcosms, and Gamma-Proteobacteria enriched in GL (but not ZDT) enrichments. Similar to Chara enrichments, the taxa VadinBC27, Mangroviflexus, Paludibacter, Barnesiella, and WCHB1-69 within the Bacteroidetes; Desulfovibrio, Desulfobacter, and Desulfomicrobium within the Delta Proteobacteria; Spirochaeta and unclassified SHA-4 within the Spirochaetes were all abundant community members at the end of enrichment. Within the Firmicutes, the family Lachnospiraceae was abundant in all enrichments, similar to what was observed in Chara microcosms. However, apart from this notable exception, the enriched community of Firmicutes genera/families differed in Chlorella microcosms when compared to Chara enrichments. Within the ZDT microcosms on Chlorella, a wide range of Clostridiales-affiliated genera and families were encountered, with members of the genera Geosporobacter (family Clostridiaceae_1), and Acidaminobacter (family Clostridiaceae_4), Youngiibacter (family Clostridiaceae_1), and members of Clostridiales Family XIII constituting ∼34% of total sequences encountered in ZDT microcosms. Further, In contrast to Chara enrichments where Veillonellaceae was only restricted to GL microcosms, Chlorella enrichments selected for members of this family in ZDT and WWT microcosms.

Chlorella enrichments selected for members of the Gamma-Proteobacteria only in GL microcosms where they constituted ∼54% of the total taxa in these enrichments. Buttiauxella and Aeromonas were identified as major taxa in GL Chlorella microcosms, similar to what was observed in Chara enrichments. In addition, members of the genus Edwardsiella (family Enterobacteriaceae) were identified as a Chlorella enrichment-specific taxon (Figs. 3–5 and Table 3). Members of the genus Edwardsiella have been repeatedly isolated from marine and freshwater animals and some species have been linked to pathogenesis in fish (Sakazaki, 1965). This is consistent with its enrichment in microcosms from a freshwater environment such as Grand Lake.

Table 3 Abundant lineages (>1%) within the abundant/enriched phyla shown in Figs. 3–5.

Phylum/Class	Class/Order	Family–genus	WWT	ZDT	GL	
Chara enrichments	
Bacteroidetes	Bacteroidales	Marinilabiaceaea-Mangroviflexus	0.58	4.16	0	
		Porphyromonadaceae-Paludibacter	1.47	1	0.1	
		Porphyromonadaceae-Bacteroides	2.11	0.05	6.91	
		Porphyromonadaceae-Barnesiella	0	0	1.58	
		Rikenellacea-VadinBC27	6.1	11.45	2.78	
	Sphingobacteriales	WCHB1-69-unclassified	0.76	3.73	0.84	
	Unclassified Bacteroidetes	14.34	4.03	0	
Firmicutes	Clostridiales	Clostridiaceae_1-Youngiibacter	0.003	1.36	0	
		Family XIII	0.41	1.85	0.1	
		Ruminococcaceae_Incertae_Sedis	0.07	0.09	1.61	
		Ruminococcaceae-Ruminococcus	0.02	0.03	2.38	
		Other Ruminococcaceae	3.62	1.47	0	
		Lachnospiraceae_Incertae_Sedis	0.16	0.45	6.87	
		Lachnospiraceae-Parasporobacterium- Sporobacterium	0	0.04	2.51	
		Veillonellaceae-uncultured	0	0	3.1	
		Unclassified Clostridiales	3.75	3.4	0.07	
	Unclassified Firmicutes	0.69	8.02	0	
Spirochaetes	Spirochaetales	Spirochaetaceae-Spirochaeta	10.67	4.69	0	
		SHA-4-unclassified	2.92	2.48	0	
	Unclassified	6.91	0.29	0	
Delta Proteobacteria	Desulfobacterales	Desulfobacteriaceae-Desulfobacter	0.36	1.39	0	
		Desulfobulbaceae-Desulfobulbous	0.28	0.69	1.22	
	Desulfovibrionales	Desulfovibrionacea-Desulfovibrio	5.69	1.42	5.67	
		Desulfovibrionaceae-Desulfomicrobium	1.89	5.69	0	
Gamma Proteobacteria	Enterobacteriales	Enterobacteriaceae-Kluyvera	0	4.15	0.04	
		Enterobacteriaceae-unclassified	0	4.2	0	
		Enterobacteriaceae-Buttiauxella	0	0	44.1	
		Enterobacteriaceae-Pantoea	0	0	11	
	Aeromonadales	Aeromonadaceae-Aeromonas	0	0.11	1.87	
Chlorella enrichments	
Bacteroidetes	Bacteroidales	Marinilabiaceaea-Mangroviflexus	0.16	2.27	0	
		Porphyromonadaceae-Paludibacter	2.59	0.46	0.81	
		Porphyromonadaceae-Barnesiella	0.002	0	1.7	
		Rikenellacea-VadinBC27	16.18	9.11	1.95	
		Other	1.78	0.2	2.95	
	Sphingobacteriales	WCHB1-69-unclassified	2.14	0.64	5.18	
	Unclassified Bacteroidetes	2.69	1.18	0	
Firmicutes	Clostridia/Clostridiales	Clostridiaceae_1-Youngiibacter	0.006	8.94	0	
		Clostridiaceae_4-Geosporobacter	0	11.44	0	
		Family_XII-Acidaminobacter	0	3.77	0	
		Family XIII	0.67	6.32	0.02	
		Lachnospiraceae_Incertae_Sedis	0.03	0.06	7.63	
		Veillonellaceae-uncultured	1.56	1.69	0.94	
		Unclassified Clostridiales	1.78	2.44	0	
	Unclassified Firmicutes	0.82	4.87	0	
Spirochaetes	Spirochaetales	Spirochaetaceae-Spirochaeta	21.65	2.62	0	
		SHA-4-unclassified	5.87	2.35	0	
	Unclassified	0.3	0.86	0	
Delta Proteobacteria	Desulfobacterales	Desulfobacteriaceae-Desulfobacter	0.03	7.31	0	
	Desulfovibrionales	Desulfovibrionacea-Desulfovibrio	2.93	1.18	21.38	
		Desulfovibrionaceae-Desulfomicrobium	0.7	6.38	0	
Gamma Proteobacteria	Enterobacteriales	Enterobacteriaceae-Buttiauxella	0	0	37.75	
		Enterobacteriaceae-Edwardsiella	0	0	7.64	
	Aeromonadales	Aeromonadaceae-Aeromonas	0	0	8.14	
Kelp enrichments	
Epsilon Proteobacteria	Campylobacterales	Campylobacteraceae-Arcobacter	8.54	0.01	0	
		Campylobacteraceae-Sulfurospirillum	2.19	0.05	0.007	
Gamma Proteobacteria	Aeromonadales	Aeromondaceae-Tolumonas	5.37	0.002	0	
	Enterobacterales	Enterobacteriaceae-Kluyvera	2.95	0.006	0	
		Enterobacteriaceae-unclassified	1.71	0	0	
	Pseudomonadales	Moraxellaceae-Acinetobacter	2.41	0	0.91	
	Other	2	0.992	5.09	
Firmicutes	Clostridiales	Clostridiaceae-Clostridium	0.49	29.55	77.73	
		Lachnospiraceae-Incertae_Sedis	0.02	19.56	0.68	
		Lachnospiraceae-Anaerosporobacter	0.004	8.1	0	
		Veillonellaceae-unclassified	65.89	0.01	0	
	Bacillales	Paenibacillaceae-Paenibacillus	0	2.1	0	

Kelp enrichments

While the microbial communities enriched on Chara and Chlorella exhibited marked similarities regardless of the inoculum source, the community enriched on kelp was quite distinct: In all kelp enrichments, Firmicutes constituted more than 70% of the total enriched taxa regardless of the inoculum source. In ZDT kelp enrichments, four different Firmicutes taxa were enriched; Clostridium, Anaerosporobacter, Lachnospiraceae-Incertae_Sedis, and Paenibacillus. Anaerosporobacter, a strictly anaerobic spore former, and other Lachnospiraceae members were previously isolated from cellulose and xylan-pectin enrichments of cow feces (Ziemer, 2014), and are frequently encountered within the human gut microbiota (Gagen et al., 2015; Lau et al., 2016; Martinez et al., 2013; Nava, Friedrichsen & Stappenbeck, 2011). Members of the genus Paenibacillus are globally distributed facultative anaerobes (Li et al., 2014), some of which are known to exhibit superior plant biomass degradation capacities (Eida et al., 2012). On the other hand, in both GL and WWT kelp microcosms, a single lineage constituted the majority of the enriched Firmicutes; Genus Clostridium in GL, and Family Veillonellaceae in WWT. Members of the genus Clostridium exhibit ubiquitious and global distribution in a wide range of anoxic habitats, while members of the family Veillonellaceae are often encountered in groundwater samples (Mosher et al., 2012), and rice paddy soil (Li et al., 2011).

In addition to Firmicutes, ZDT kelp microcosms showed an abundance of the Spirochaetes genus Treponema (20.2% of the total enriched taxa) previously shown to contribute to the overall cellulolytic activities in barley straw microcosms (Kudo, Cheng & Costerton, 1987), and WWT kelp microcosms showed an abundance of members of Epsilon (genera Arcobacter and Sulfurospirilum) and Gamma (genera Tolumonas, Kluyvera, and Acinetobacter) Proteobacteria, collectively comprising ∼25% of the total enriched taxa. Members of these genera were previously implicated in anaerobic plant biomass degradation (Billings et al., 2015; Caldwell et al., 2011; Cardoso et al., 2012).

Discussion

In this study we investigated the microbial community mediating algal detritus turnover under anaerobic conditions. We utilized three representative algal species: Chlorella vulgaris strain UTEX2714 representing the Chlorophyta, Chara sp. strain IWP representing the Charophyceae, and Ascophyllum nodosum (kelp) representing the brown algae (Phaeophyceae). We followed the turnover of these algae in enrichments that were set up using three different sources of inoculum: an anoxic freshwater sulfide- and sulfur-rich spring (Zodletone spring, OK), a wastewater treatment plant (Municipal wastewater treatment plant in Stillwater, OK), and a seasonally stratified lake that experience seasonal algal blooms (Grand Lake O’ the Cherokees, OK). We identified multiple microbial lineages that were significantly enriched in these treatments. Some of these lineages appear to be substrate-specific (i.e., enriched when using a specific algal species as a substrate source regardless of the inoculum source utilized, e.g., VadinBC27 that was enriched on Chara and Chlorella regardless of the inoculum source and Spirochaeta that was enriched on Chara and Chlorella in ZDT and WWT microcosms), habitat-specific (i.e., enriched only when using a specific source of inoculum regardless of the algal substrate utilized, e.g., Buttiauxella, that was enriched in GL microcosms regardless of the algal substrate), or treatment-specific (i.e., encountered only in a specific algal substrate/inoculum source combination, e.g., Arcobacter in WWT microcosms on kelp, Geosporobacter, Acidaminobacter, Anaerosporobacter, and Treponema in ZDT microcosms on kelp, Youngiibacter in ZDT microcosms on Chlorella, and Pantoea in GL microcosms on Chara).

Within all nine treatments examined, a high level of diversity was invariably retained at the conclusion of the incubation process. We reason that this is a reflection of the complexity of the substrate utilized. Algal detritus harbors multiple complex macromolecules, e.g., proteins, lipids, nucleic acids, and polysaccharide, that vary considerably in structure and hence require multiple enzymes and pathways for their efficient degradation (e.g., pectin and cellulose in algal cell walls require an arsenal of degradation enzymes (Abbott & Boraston, 2008; Doi & Kosugi, 2004)). Such level of complexity could potentially select for a wide range of organisms, each contributing to the degradation process of a specific substrate within the algal biomass. This is in stark contrast to the selection of one/few microbial lineages in anaerobic incubations conducted using a single, chemically defined substrate (Viggor et al., 2013; Yagi et al., 2010).

Our results and subsequent community analysis (Figs. 2–5 and Table 3) indicate that kelp enriched for a highly similar microbial community that is mostly composed of members of the order Clostridiales; genus Clostridium and Anaerosporobacter and family Veillonellaceae, regardless of the inoculum source (ZDT, WWT, and GL). While only a handful of environments were examined in this study, the consistent selection for members of a specific lineage regardless of the starting inoculum suggests the ecological significance of this lineage in kelp detritus turnover in anaerobic habitats. The reason for this observed pattern of Clostridiales genera/families selection on kelp could only be speculated upon. A possible contributing factor could be the unique cell wall structure of kelp (or brown algae); multiple cellulose microfibrils layers embedded in large interfibrillar matrices that are mostly composed of alginates and fucans (Domozych, 2001; Youssef et al., 2015). Alginate (Preiss & Ashwell, 1962a; Preiss & Ashwell, 1962b) and fucans (Descamps et al., 2006; Kusaykin et al., 2016) degradation requires highly specific enzymes machineries. Organisms with alginate or fucan/fucoidan-degradation capabilities under aerobic conditions have been previously isolated (Ekborg et al., 2005; Jagtap et al., 2014; Park et al., 2012; Sakai, Kawai & Kato, 2004; Thomas et al., 2012; Yonemoto et al., 1993). On the other hand, with the exception of a few studies that used anaerobic batch-fed mixed inocula to degrade brown algae and produce methane (Moen, Horn & Østgaard, 1997a; Moen, Horn & Østgaard, 1997b; Sutherland & Varela, 2014), there is a scarcity of information on the identity of the degrading inocula under anaerobic conditions. In contrast to the number of studies on the anaerobic degradation of other common polysaccharides, e.g., cellulose and xylans, a single study by Kita et al. (2016) reported on the identity of a bacterial consortium (formed mainly of a Clostridiaceae bacterium and a Porphyromonadaceae bacterium (Dysgonomonas capnocytophagoides)) anaerobically degrading alginate. Based on the study by Kita et al. (2016) and the results we report here, it is possible that members of the Clostridiales represent one of very few members possessing alginate and/or fucan-degrading capabilities and that are readily enriched and propagated under laboratory incubations.

On the other hand, when using Chara or Chlorella as an algal inoculum, the final microbial community enriched was highly divergent, and the final community structure was mostly dependent on the inoculum sources (ZDT, WWT, GL), rather than the type of algal substrate provided (Figs. 2–5, Table 3). While Chlorella and Chara cell walls are quite distinct, they are both similar in being rich in fibrous cellulose and/or hemicellulose with amorphous middle layers composed mainly of pectin (homogalacturonic and rhamnogalacturonic acids polymers) in Charophyta, or sulfated polysaccharides in Chlorophyceae (Domozych, 2001; Domozych et al., 2014; Youssef et al., 2015). We reason that the relative similarity of the communities enriched on both types of algae, as well as the enrichment for multiple, rather than a single group of microbial lineages (VadinBC27, Spirochaeta, Lachnospiraceae, Buttauxiella, and Pantoea) is a reflection of the relative ubiquity of microbial lineages capable of the anaerobic degradation of cellulose, hemicellulose, and pectin in the algal cell walls, hence allowing ready access to the intracellular substrates within the algal cells.

Under anaerobic conditions, multiple groups of organisms and metabolic guilds are often required for the effective and complete degradation of complex organic molecules (McInerney, Sieber & Gunsalus, 2009; Morris et al., 2013). By examining the known metabolic capabilities of close relatives of lineages enriched in various treatments, one could propose a model depicting their putative involvement in the complex algal detritus degradation processes. In kelp enrichments, complex carbohydrate polymer degradation to monomers could possibly be mediated by various members of the order Clostridiales (Clostridium, Anaerosporobacter, Lachnospiraceae incertae sedis) as shown before (Ziemer, 2014), as well as the fermentative bacteria in the Enterobacteriaceae (Kluyvera) (Xin & He, 2013). Produced sugar monomers can be further fermented to various fatty acids (acetate and longer chain fatty acids e.g., butyrate, propionate, etc.) by the same members of the Clostridiales and Enterobacteriaceae, as well as the Epsilon Proteobacterium Sulfurospirillum (Stolz et al., 1999). Proteins in the initial substrate could potentially be degraded by the Epsilon Proteobacterium Arcobacter (Roalkvam et al., 2015). Additionally, while the majority of sequences obtained were bacterial in origin, the few archaeal sequences obtained suggest the enrichment of members of Bathyarchaeota (Table S2). Previous research using genomic sequences of different members of the Bathyarchaeota suggested their involvement in both complex carbohydrates and detrital protein degradation as well as acetate production (Lazar et al., 2016), which could explain their enrichment on kelp. Under anaerobic condition, syntrophic organisms convert the long chain fatty acids produced from the initial polymer degradation to acetate. Definitive identification of syntrophic organisms in culture-independent studies is challenging, given their close phylogenetic affiliation with fermentative lineages (Morris et al., 2013). On the other hand, saccharolytic clostridia members of the family Lachnospiraceae could potentially perform the initial breakdown of polymeric substances and the fermentation of the resulting sugars to acetate, hydrogen, and CO2 (Krumholz & Bryant, 1986). While other obligate syntrophic organisms, e.g., members of the families Syntrophobacteraceae, Syntrophaceae, Syntrophomonadaceae, and Syntrophorhabdaceae, were detected in very low percentage (<0.06% of the total community in any enrichment), their role could not be ruled out. The produced acetate, hydrogen, and CO2 would eventually be converted to methane by methanogens. The role of methanogens as the dominant terminal electron acceptor in kelp enrichment from ZDT and WWT inoculum sources is suggested by the observed increase in mrcA gene copy number in qPCR analysis (Fig. 1) and the identification of several sequences affiliated with known methanogens (genera Methanosarcina, Methanothermococcus, Methanogenium, and Methanomicrobium) in kelp enrichment from ZDT (Table S2). The lack of sulfate utilization in all kelp enrichments (Fig. S2) argues against the involvement of the SRBs identified in the culture-independent dataset (Desulfovibrio, Desulfobacter, Desulfomicrobium, and Desulfobulbous) and detected by qPCR (Fig. 1) in the process. Similar results were previously shown in biofilms (Santegoeds et al., 1998), where not all SRBs detected by culture-independent techniques were found to be sulfidogenically active.

Within Chara and Chlorella enrichments, complex carbohydrate (e.g., cellulose, pectin, hemicellulose) degradation to sugar monomers could be mediated by members of the Bacteroidetes uncultured groups VadinBC27 (in all enrichments from all sources) and WCHB1-69 (in Chara enrichment from ZDT, and Chlorella enrichment from WWT and GL), as well as the Spirochaetes (Gao, Xu & Ruan, 2014) (genus Spirochaeta and the uncultured group SHA-4 enriched on Chara and Chlorella from ZDT and WWT sources). These lineages have been consistently enriched in anaerobic sludge digestors (Godon et al., 1997; Lee et al., 2013), and microcosms with hydrocarbon or halogenated solvents (Dojka et al., 1998; Gu et al., 2004; Xu et al., 2012). Similarly, members of the Clostridiales (Family Ruminococcaceae (in Chara enrichments from all sources), Family Veillonellaceaea (in Chara enrichments from GL), Family Lachnospiraceae (in Chara and Chlorella enrichments from GL), and Family Clostridiaceae genera Yongiibacter, Geosporobacter, Acidaminobacter (in Chloreela enrichments from ZDT)), as well as Enterobacteriales (Genera Kluyvera (Chara enrichments from ZDT), Pantoea (Chara enrichments from GL), Edwardsiella and Aeromonas (Chlorella enrichment from GL), and Buttiauxella (Chara and Chlorella enrichments from GL)) could potentially mediate complex carbohydrate degradation (Hegler et al., 2012; Jiménez, Chaves-Moreno & Van Elsas, 2015; Jiménez et al., 2016; Sakazaki, 1965; Wust, Horn & Drake, 2011; Xin & He, 2013; Ziemer, 2014). The monomers produced could potentially be converted to long chain volatile fatty acids, acetate, and H2 by the Clostridiales and Enterobacteriales members above. Alternatively, long chain volatile fatty acids could be converted to acetate, H2 and CO2 by syntrophs, or oxidized either completely (to H2 and CO2) or incompletely (to acetate, H2 and CO2) by sulfate-reducing bacteria (e.g., the complete oxidizers (Desulfobacter in Chara and Chlorella enrichments from ZDT), or the incomplete oxidizers (Desulfovibrio in all Chara and Chlorella enrichments from all sources, Desulfomicrobium in Chara and Chlorella enrichments from ZDT and Chara enrichments from WWT, and Desulfobulbous in Chara enrichment from GL)) when sulfate is available. The produced acetate, H2 and CO2 could either be metabolized to methane by aceticlastic or hydrogenotrophic methanogenic lineages observed in the enrichments (Table S2) (e.g., the aceticlastic Methanosarcina in Chara and Chlorella enrichments from ZDT and WWT, and the hydrogenotrophic Methanothermococcus in Chlorella early (week 7) enrichments from ZDT), or metabolized by the aceticlastic autotrophic SRBs in the presence of sulfate. The increase in dsr copy numbers in Chara and Chlorella enrichments as measured by qPCR, the utilization of the available substrates in these enrichments (loss of sulfate (Fig. S2)), as well as the presence of a large and diverse community of SRBs (Table 3 and Table S2) evidenced by the culture-independent analysis, strongly argue for the co-involvement of sulfate reduction and methanogenesis as two competing terminal electron accepting processes in these enrichments. Recently, the methanogenic potential for members of the Bathyarchaeota was suggested based on genomic metabolic reconstruction (Evans et al., 2015). It is worth noting that the Bathyarchaeota phylum was enriched in ZDT Chara and Chlorella microcosms (Table S2) and could potentially be contributing to methanogenesis in these enrichments.

In conclusion, our work represents the first systematic survey of microbial communities mediating turnover of algal biomass under anaerobic conditions, and highlights the diversity of lineages putatively involved in the degradation process. The results presented here could certainly open the door for future studies that investigate the interactions between the abundant genera identified as significant for the degradation process, as well as for targeted isolation studies for algal detritus degraders.

Supplemental Information

Supplemental Information 1 Supplementary text

Includes two supplementary tables and two supplementary figures

Table S1

Table S2

Figure S1

Figure S2

Click here for additional data file.

Additional Information and Declarations

Competing Interests

Author Contributions

DNA Deposition

Data Availability

The authors declare there are no competing interests.

Richard M. Zamor and Steve Nikolai are employees of Grand River Dam Authority (GRDA).

Jessica M. Morrison performed the experiments, analyzed the data, wrote the paper, prepared figures and/or tables, reviewed drafts of the paper.

Chelsea L. Murphy performed the experiments, analyzed the data, reviewed drafts of the paper.

Kristina Baker performed the experiments, reviewed drafts of the paper.

Richard M. Zamor and Steve J. Nikolai contributed reagents/materials/analysis tools, reviewed drafts of the paper.

Shawn Wilder performed the experiments, contributed reagents/materials/analysis tools, reviewed drafts of the paper.

Mostafa S. Elshahed conceived and designed the experiments, contributed reagents/materials/analysis tools, wrote the paper, reviewed drafts of the paper.

Noha H. Youssef conceived and designed the experiments, analyzed the data, contributed reagents/materials/analysis tools, wrote the paper, prepared figures and/or tables, reviewed drafts of the paper.

The following information was supplied regarding the deposition of DNA sequences:

Genbank accession number SRP083898.

The following information was supplied regarding data availability:

The research in this article did not generate, collect or analyse any raw data or code.

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
