# Peer review of "Microbial communities mediating algal detritus turnover under anaerobic conditions"

_PeerJ, doi:10.7717/peerj.2803_

## Round 0.1 · original submission · Minor Revisions

Your manuscript was reviewed by three referees. Each identified strengths of the manuscript and overall were generally favorable for eventual publication. However, each also identified multiple areas where aspects of the manuscript and the work described could use modification. I believe their comments are valid and important to consider.

·

Basic reporting

The manuscript passes all criteria as far as I was able to assess.

Experimental design

Regarding the knowledge gap, these are the relevant sentences from the ms:

A surprising lack of knowledge exists regarding the microbial community and patterns of algal turnover under the anoxic conditions in the lower layers of stratified water bodies.

While various aspects of the engineering and performance have been studied, there is very little documentation of the identity of the microbial community that is mediating algal detritus turnover under these anaerobic conditions.

To our knowledge, this is the first study that systematically characterized the microbial community associated with algal degradation under anaerobic conditions.

The study was carefully designed to help fill this gap.

Validity of the findings

I could not find statements in the text indicating which public repository if any contains the study's data.

The manuscript complies with the remaining items.

Additional comments

I found this study to be well-designed, executed, and analyzed. The one criticism I have is that the study is entirely based on the V4 region of the 16S gene. While on the one hand this is standard technique for microbial community profiling, on the other hand it suffers from biases that may result in a partial view of the true composition of any one environment. It's becoming more common to deal with this shortcoming by supplementing the 16S approach with total DNA shotgun sequencing.

On this topic, I found the following result strange, and deserving some explanation. In Table S2, unclassified OTUs for bacteria have relatively high percentages for ZZT and WWT, but not for GL. Why? Whatever the answer, I suggest that these results (that there are significant fractions of unclassified sequences) be highlighted in the main text.

I understand what is the meaning of OTU 0.1 but it is never defined.

line 180: surficial => superficial

Reviewer 2 ·

Basic reporting

The article is overall well written and has a clear and appropriate structure. The introduction gives a sufficient and comprehensive background. The introduction use in same instances overly emotional language (e.g. line 82 “bewildering”, line 98 “fantastic”, line 112 “surprising”), which could be omitted. The figures are of good standard and appropriate. All relevant data is either available through the submission or through public database.

Experimental design

The work is original as it investigates a knowledge gap in understanding microbial processes involved in the degradation of algal material. This leads to a well-justified and clear aim for the submitted study.
The methods are generally performed to an appropriate high standard, however there are some issues that need to be addressed:

1) In line 155, kelp powder is stated to act as an inoculum for the experiments, but it should rather be described as a carbon source. Also a short comment why kelp powder was used and not fresh biomass as done for the other two algae investigated. The description of the powder should also be more specific (e.g. who is the local provider?) and also should consider whether indigenous microbes might be present that could contribute to degradation (note that the other two algae would have likely contained some microbes that could proliferate during enrichment).
2) All treatments were set up in triplicate and then sampled over time. However only one data point is present for each time point and treatment for the 16S rRNA gene sequence analysis (see Figure 2). Please explain (and justify), if this one data point is derived from one random sample or from all triplicates being pooled. Also consider highlighting time point in Figure 2 and ensure that the symbols in the figure legend match the ones in the plot (they did not in the copy I had)
3) One set of samples was set up without any algal addition (Line 203-204), but it unclear when it was analysed and what kind of analysis was done (qPCR?). In the following text these samples are defined as “pre-enrichment” samples, which would imply that they were analysed straight after being set up. Please clarify this. I also think that such a treatment control (i.e. no algae) should be analysed over time to define which microbes in the inoculum change in abundance over time simply by being kept in the basal medium.
4) The description of the 16S rRNA sequencing data with MOTHUR reads very similar to the SOP published by the program developers. If this is true, then appropriate reference should be made.
5) Where 16S rRNA gene sequencing data sub-sampled, normalised or singleton-OTUs removed prior to alpha and beta-diversity analysis? Please state or justify what this was not done (or done).
6) Please describe how dry weight was measured.

Validity of the findings

The results and conclusion are well presented, but could be more rigorously analysed. The following would strengthen the manuscript:
1) The qPCR results appear to be replicated within each time point and treatment (as inferred from the error bars in Figure 1). This should allow for a statistical test (GLM) to support the claim that cell numbers of specific groups indeed increase (or not).
2) While there might be no replication for the 16S rRNA gene community data (see comment above), there seem to be trends over time (Figure 3) that could be analysed via statistical correlation analysis.
3) The description of the results mostly focuses on OTUs that could be taxonomically classified. What happened to OTUs that could not be classified? Or was the proportion of OTUs without taxon assignment very small/ neglible? Please comment on this.
4) Please note that there have been recent findings that the Bathyarchaeota can perform methanogenesis (e.g. Evans et al. Science 2015). This should be considered in the interpretation of the data.

Additional comments

Some minor points:
Lines 161-162: Consider rephrasing “oxygen levels in deeper layers becoming completely anoxic” to perhaps “deeper layers becoming anoxic”

Line 206: The sentence “All treatments..” is redundant as the level replication is mentioned before.

·

Basic reporting

The current article meets the basic reporting criteria. The authors provide adequate background and rationale for the study and the subsequent design, execution and results of the study speaks to the overall study.
Line 75, remove aquatic
Line 82, replace experience with exhibit
Line 98, replace algal growth with algae

Experimental design

Degradation of different algal detritus by anaerobic microbial communities from three different inocula. Microbial community of each inoculum determined before, during and after experimental period. Changes in the amount of organic and inorganic materials, as well as bacterial load determined during and after.
Overall, the study was was well designed and executed.There are however, some minor concerns.

Validity of the findings

The results of the study were valid and insightful. The conclusions drawn were appropriate. There are however, some minor concerns that should be addressed.

1. In the data analyses, the authors used a cut of 5% for phyla they considered important to the degradation process based on phyla that were abundant before, during and after the study, as well as those that increased transiently during parts of the study. And subsequently, only genera that made up 1% or more of the total abundance. This approach enabled the authors focus on taxa they considered important to the degradation process.

However, the authors acknowledge the importance of potential syntrophy between organism that enables the degradation of polymers in the enrichments. In as much as definitive identification of these synthrophic organisms might be difficult, perhaps analysis of the community composition of the taxa that fail to make the 5% cut-off threshold may be informative. Several instances in the discussion are attributed to synthrophy (sulphate reduction and methanogenesis, as well as the importance of multiple enzyme pathways for efficient degradation of organic matter), indicative of potential metabolic connections between the most abundant and least abundant bacterial taxa in these enrichments. Perhaps a mention of some cursory analysis of the reads below the 5% and 1% thresholds may be informative. Not required for the current submission, but would be insightful and provide some support for the proposed synthrophic interactions between microbes in the enrichments.

2. The authors should redo the figure 1 legends and the figure 1. The bars do not correspond to the legend and its hard to determine which groups are increasing or decreasing. Maybe three different panels for each inocula may allow for easier assessment? Eg. Fig. 1A for WWT inoculum, Fig. 1B, for GL inoculum..etc

3. In the enrichment setup, the anaerobic medium was generated as described and cysteine-HCL added as a reductant and resazurin used as redox indicator. Was Nitrogen, Hydrogen or a mix of these gases flushed through the medium to make it anaerobic (indicated by change in color by resazurin) before the medium was autoclaved, or was the medium autoclaved without this process?
The reason for this question is that in addition to anaerobic taxa in these enrichments, several of the microbial lineages in the final communities in the enrichment were also facultative anaerobic bacteria. Perhaps the absence of some of the microbial taxa implicated in the suggested synthrophic relationships may be due to inadequately anaerobic conditions?

4. The authors should highlight some future studies stemming from the results of this study, that might investigate or assess some of the possible synthrophic interactions related to algal detritus degradation.

Additional comments

Overall, i am of the opinion this manuscript meets the standards set by PeerJ and that the research itself is pretty impressive as it relates to biofuel production. My comments regarding the generation of the anaerobic medium, as well as the analysis of the least abundant taxa i believe will solidify the manuscript when addressed.

---

## Round 0.2 · Minor Revisions

All reviewers believe you have addressed sufficiently the comments on the previous version. However, one reviewer has noted a minor problem with Figure 2 that should be addressed.

·

Basic reporting

The authors have responded to all points raised to my satisfaction.

Experimental design

The authors have responded to all points raised to my satisfaction.

Validity of the findings

The authors have responded to all points raised to my satisfaction.

Reviewer 2 ·

Basic reporting

All my previous concerns have been addressed.

Experimental design

All my previous concerns have been addressed.

Validity of the findings

All my previous concerns have been addressed.

Additional comments

Please edit Figure 2 to make sure that the new labels overlap with the old labels completely. Currently is looks quite messy.

·

Basic reporting

Meets standards

Experimental design

Meets standards

Validity of the findings

Meets standards

---

## Round 0.3 · accepted · Accept

All looks acceptable now.